# Production of Vegetables and Artichokes Is Associated with Lower Cardiovascular Mortality: An Ecological Study

**DOI:** 10.3390/ijerph17186583

**Published:** 2020-09-10

**Authors:** Alberto Arnedo-Pena, Joan Puig-Barberà, Juan Bellido-Blasco, MªAngeles Romeu-Garcia, Mª Rosario Pac-Sa, Francisco Guillen-Grima

**Affiliations:** 1Health Sciences Department, Public University Navarra, 31008 Pamplona, Spain; f.guillen.grima@gmail.com; 2Epidemiology Division Public Health Center, 12003 Castello de la Plana, Spain; bellido_jua@gva.es (J.B.-B.); romeu_man@gva.es (M.R.-G.); charopac@gmail.com (M.R.P.-S.); 3Epidemiology and Public Health (CIBERESP), 28029 Madrid, Spain; 4Vaccines Research Area FISABIO, 46020 Valencia, Spain; jpuigb55@gmail.com; 5Epidemiology Department, Jaume I University, 12006 Castelló de la Plana, Spain

**Keywords:** mortality, cardiovascular, cerebrovascular, vegetables, artichokes, agriculture, municipalities, multilevel, ecology

## Abstract

Mortality due to cardiovascular disease (CVD), including cerebrovascular disease (CED) and ischaemic heart disease (IHD), was considerably different in eight municipalities of the province of Castellón, Community of Valencia (Spain) during the period of 1991–2011. In addition, these villages showed differences in agricultural practices and production. Since high vegetable consumption has been linked to decreased all-cause, CVD, and CED mortalities, we hypothesized that the diversity in vegetable and artichoke production, used as proxies for their consumption, could be associated with the diversity of mortality rates. In order to test our hypothesis, we estimated the smoothed standardized mortality ratios (SMRs) of CVD, CED, and IHD mortalities and a directed, age-adjusted mortality rate (AMR). We used a multilevel linear regression analysis to account for the ecological nature of our study. After adjustment, the CVD and CED SMRs were inversely associated with vegetable and artichoke production, with a reduction in SMRs for CVD: −0.19 (95% Confidence Interval [CI] −0.31 to −0.07) and −0.42 (95% CI −0.70 to −0.15) per hectare/10^3^ inhabitants, respectively. The SMRs for CED also decreased: −0.68 (95% CI −1.61 to −0.19) and −1.47 (95% CI −2.57 to −0.36) per hectare/10^3^ inhabitants, respectively. The SMRs for IHD were not associated with vegetal and artichoke production. When the directed AMR was used, CED mortality was consistent with the previous results, whereas the CVD mortality association was lost. Our results indicate that vegetable and artichoke production may act as protective factors of CED and CVD mortalities.

## 1. Introduction

Mortality from cardiovascular diseases (CVD), including cerebrovascular disease (CED), is the primary cause of death in the population, and many risk factors have been identified through epidemiological and medical studies [1,2,3,4,5]. CVD and CED mortalities present large geographic differences, suggesting that factors at a local level may play a role in the observed variability. To explain these differences, research on mortality in small areas is useful to generate hypotheses on potential causes or risk factors, health resource distribution, and preventive measures [6,7,8]. In general, the first stage is the detection of local differences in CVD mortality. If differences are found, a second stage is carried out to estimate risks and protective factors [9,10,11,12].

Using the Analysis Geographic Epidemiologic (AGEPI) mortality program [13,14,15], we observed differences in CVD and CED mortalities in the eight municipalities with 25,000 or more inhabitants in the Castellón province in the Community of Valencia (CV); today a new version is available [16]. There is a difference between these municipalities in their agricultural practice: citrus cultivation is predominant in all of the municipalities except one, Benicarló, where vegetables, mainly artichokes (*Cynara scolymus*), have high production levels. Consumption of vegetables and fruits is associated with lower all-cause, CVD, CED, and ischaemic heart disease (IHD) mortality [17,18,19]. Artichokes have properties, such as lipid and glycemic reduction and high levels of K with a low Na/K index, that may prevent hypertension and CVD [20,21].

Our hypothesis was that the production of vegetables and artichokes as a proxy of their consumption is associated with lower CVD and CED mortalities. To test this, we conducted an ecological study considering all-cause, CVD, CED, and IHD mortalities in eight municipalities in the Castellón province.

## 2. Materials and Methods

We conducted an ecological study considering all-cause, CVD, CED, and IHD mortalities during the period from 1991 to 2011 in the eight municipalities in the province of Castellón with a population of 25,000 or more inhabitants in 2011. This province is divided into three health departments: Vinaròs in the north, Castelló de la Plana in the center, and Vila-real in the south, with 90,816, 283,021 and 187,258 inhabitants, respectively.

The municipalities were Almassora (25,945 inhabitants), Benicarló (26,553 inhabitants), Borriana (27,385 inhabitants), Castelló de la Plana (176,300 inhabitants), Onda (25,704 inhabitants), la Vall d’Uixó (32,864 inhabitants), Vila-real (51,168 inhabitants), and Vinaròs (28,508 inhabitants). The health care system was the same in the 8 studied municipalities, and the CVD medical attention was comparable in the three health departments (Vinaròs, Castelló de la Plana, and Vila-real) with a ratio of 1299, 1497, and 1538 physicians per 1000 inhabitants, respectively, and a ratio of 1613, 1756, and 2064 nurses per 1000 inhabitants, respectively [22].

The province of Castellón is located on Spain’s east, Mediterranean coast, with 576,889 inhabitants and an area of 6612 km^2^. The population is concentrated in the north (El Baix-Maestrat), in Benicarló and Vinaròs, in the central part (La Plana Alta), in the cities of Castelló de la Plana, the administrative capital, and Almassora, as well as in the southern part (Plana Baixa), in Borriana, Vila-real, la Vall d’Uixó, and Onda. Almassora and Vila-real are located adjacent to Castelló de la Plana, 5.2 km and 7.6 km apart, respectively, and Borriana is located 11.1 km south of Castelló; all of these municipalities are on or near the Mediterranean coast. In the interior, la Vall d’Uixó and Onda are located 24.5 km and 19.3 km from Castelló de la Plana, respectively. Vinaròs and Benicarló are near each other (7.2 km apart) and are located 70 km and 62.3 km away from Castelló de la Plana, respectively (Figure 1). Agricultural production is based on citrus fruits in all of these municipalities, with the exception of Benicarló, where vegetable production predominates with artichokes as the more prevalent vegetable. Industrial production involving ceramics is located in Vila-real and Onda, and the petro-chemistry industry is based in Castelló.

The number of CVD, CED, and IHD deaths was obtained from the AGEPI mortality program of the Conselleria de Sanitat Universal and Salud Pública of the Generalitat Valenciana [13]. The causes of mortality were coded according to the 9th and 10th International Classification of Diseases (ICD): CVD (codes 100–199; 390–459); CED (codes 160–169; 430–438); IHD (codes 120–125; 410–419) [23].

We obtained socio-economic and demographic data from the Generalitat Valenciana (Argos, Banc de dates Territorial) and from the National Statistics Institute (2001–2011 censuses) [24,25,26]. We collected information on the number of inhabitants, sex distribution, population aged 65 years and above, foreign-born population, household income in euros, unemployment, illiterate population, percentages of occupations in agriculture, industry, services, and construction as variables for adjustment, and agricultural production of vegetables, artichokes, and citrus fruits per hectare per 1000 inhabitants (median for the period 2002–2011) as explicative variables.

## 3. Statistical Methods

We estimated the smoothed standardized mortality ratios (SMRs) using Clayton and Kaldor’s methodology [27] and compared the observed mortality with the expected mortality in the CV, and calculated the age-adjusted mortality rate (AMR) by direct standardization of the CV age distribution per 100,000 inhabitants during the study period [13]. The SMRs were calculated from each of the 8 municipalities’ observed mortality and the expected mortality from Valencia’s population. Given that calculating the SMR is a preferred method to study mortality in small areas [28], we used AMR calculation as a sensitivity analysis for the SMR results, considering direction and significance. Using the Directed Acyclic Graphic (DAG) [29] (Figure 2), we studied the relationship of exposure to agricultural production with all-cause, CVD, CED, and IHD mortality outcomes, and potential confounding factors: sex, population aged 65 years and above, foreign-born, household income, illiteracy, and municipality. Other potential confounders such as unemployment, particulate matter (PM10) and nitrogen dioxide (NO2) levels, coastal proximity, and drinking water hardness were studied.

We used a multilevel linear regression analysis with municipality as the level variable to estimate associations between mortality outcomes and exposure variables in models adjusted for potential confounders assuming normal data distribution. A residual interclass correlation (ICC) indicated the total explained variance by the models. We used Log likelihood, Akaike’s information criterion and Bayesian information criterion to check the goodness of fit of the models. We estimated the Moran’s I index [30] to evaluate the spatial autocorrelations of the SMRs and AMR among the municipalities and used Spearman’s nonparametric correlation to detect relationships among exposure variables. We performed all of the analyses with Stata^®^ version 14 (StataCorp, College Station, TX, USA. No Investigation Review Board approbation was required because all data were anonymous-aggregated data of public domain.

## 4. Results

The socio-economic characteristics of the eight municipalities (Table 1) indicated some important differences. Castelló de la Plana, the province capital, was the biggest municipality, and Almassora the smallest. The foreign-born population was higher in Benicarló and lower in la Vall d’Uixó. Household income differences were small, but unemployment was higher in Onda and lower in Benicarló and Vinaròs. Industrial occupations were more common in Onda and Vila-real. Agricultural hectares per inhabitant were higher in Onda and Vinaròs and lower in Castelló. A great contrast between Benicarló and the other cities was observed when vegetable and artichoke production levels were compared. Benicarló had the highest level of vegetable and artichoke production, whereas in Onda and Vila-real, vegetable and artichoke production were almost null. Despite high citrus production in the province, differences were considerable; Borriana and Vinaròs had the highest production level and Castelló the lowest. Out of the eight municipalities, except Onda and Vall d’Uixó, six were located in coastal proximity, and all of the municipalities had a Mediterranean climate, although rains were more present in Benicarló and Vinaròs. Water hardness was elevated in these municipalities, with higher levels in Castelló and the lowest in Vinaròs. Air pollution, measured by PM10 and NO2 levels, was higher in Almassora and lower in Benicarló and Vinaròs.

Spearman correlation coefficients among explicative variables are shown in Table 2. Vegetable and artichoke production levels had strong positive correlations (0.91); artichoke production correlated negatively with unemployment (−0.92) and positively with coastal distance (0.87). A strong negative correlation between foreign-born population percentage and unemployment was found (−0.83). Pm10 and NO2 negatively correlated with the percentage of population aged 65 years and above and agricultural and vegetable production.

The SMRs for all-cause and CVD mortalities are presented in Table 3. For males, the SMR for all-cause mortality was higher than 100 in all cities except Benicarló. For CVD, CED, IHD, and CVD plus IHD mortalities, Benicarló had the lowest SMRs and Borriana the highest. Benicarló had SMRs lower than 100 for all of these diseases. For females, as with males, Benicarló had the lowest SMRs for all-cause and CVD mortalities. We observed higher differences across the municipalities in the case of females: Onda had the highest SMR for all-cause mortality, Vila-real had the highest for CVD, Borriana for CED, and Castelló for IHD.

The AMRs for all-cause and cardiovascular disease mortalities are presented in Table 4. The results are parallel with the SMRs. For males, Benicarló had the lowest AMRs for all-cause and CVD mortalities, and Borriana the highest. The AMR differences across the municipalities are elevated, being the highest for CED and the lowest for IHD. For females, Benicarló had the lowest AMRs for all-cause and CVD mortalities; Onda had the highest AMRs for all-cause and CVD mortalities, followed by Borriana in CVD mortality and Castelló in IHD mortality.

The spatial autocorrelation on SMRs and AMRs was significant for CED plus IHD. CVD was only found to be significant on SMRs (Table 5).

With the SMRs and AMRs as dependent variables, and agricultural, vegetable, artichoke, and citrus production as independent variables adjusted for other covariates (Table 6), artichoke production was associated with lower SMRs or AMRs in all instances.

All-cause mortality was inversely associated with vegetable and artichoke production, but ICCs were not computable; we found no association between agricultural and citrus production with all-cause mortality. CVD mortality (SMR) was reduced: −0.19 (95% CI −0.31 to −0.07) for vegetables and −0.42 (95% CI −0.79 to −0.04) for artichokes per hectare/10^3^ inhabitants, with low ICCs of 0.07 and 0.10, respectively. For CEV, the SMRs were lower: −0.68 (95% CI −1.16 to −0.19) for vegetables and −1.4 (95% CI −2.57 to −0.36) for artichokes per hectare/10^3^ inhabitants, with high ICCs of 0.72 and 0.73, respectively. Agricultural and citrus production levels were not associated with CVD or CED mortalities (SMRs). IHD mortality (SMRs) was not associated with any of the independent variables and ICCs were not computable. For CED plus IHD mortality, when vegetable and artichoke production increased, the SMRs decreased, with ICCs of 0.59 and 0.61, respectively.

The AMR multilevel linear regression sensitivity analysis results are reported in Table 6. The AMR results closely resemble those obtained for the SMRs. All causes of mortality were inversely associated with vegetable and artichoke production, but the ICCs were not computable, and agricultural and citrus production levels were not associated with all-cause mortality. CVD mortality was not associated with any of the independent variables. CED mortality was inversely associated with vegetable and artichoke production with ICCs of 0.39 and 0.41, respectively; agricultural and citrus production levels were not associated with any change in CED mortality. IHD mortality was inversely associated with vegetable and artichoke production but ICCs were not computable; agricultural and citrus production levels were not associated with IHD mortality. CED plus IHD mortality was inversely associated with vegetable and artichoke production with ICCs of 0.56 and 0.58, respectively. Agricultural and citrus production levels were not associated with CED plus IHD mortality.

## 5. Discussion

Our results support that the production of vegetables and artichokes may be protective for CED mortality and, to a lesser extent, for CVD mortality. A non-significant effect was observed in IHD mortality. All-cause mortality was inversely associated with the production of vegetables and artichokes, but with a low ICC. By contrast, CVD, CED and IHD mortalities were not associated with agricultural and citrus production, suggesting some specificity for the estimated protective effects of vegetables and artichokes.

In the sensitivity analyses of the AMRs, we obtained results similar to the SMRs in respect to CED and all-cause mortalities, but the CVD mortality association with vegetable and artichoke production was not significant. The autocorrelation did not indicate a general spatial dependence and other municipality-related factors could have played a role. Our findings support that SMRs and AMRs may be adequate for spatial analysis of CED mortality.

CVD is related to many risks and protecting factors, including socio-economic and personal factors, such as lifestyle, social class, diet, stress, air pollution, health, genetic conditions, and health care systems [31,32,33,34,35,36]. In our study, Borriana presented high all-cause, CVD, and CED mortalities; this situation may be related with the elevated prevalence of type 2 diabetes mellitus and obesity reported there in a 2006 publication [37]. All-cause, CVD, and CED mortalities have been associated with blue water and green space [38,39]. In addition, water hardness and CVD mortality’s relationship has been studied for a long time but controversy remains [40,41]. The hardness of drinking water (calcium and magnesium concentrations) was associated with CED and IHD mortalities in a study of CVD at a municipal level in the CV [9]; when the level of magnesium was controlled in the models, the SMRs of CED mortality increased, which suggested a protecting effect apart from other unknown factors. Interestingly, psychosocial stress has been shown as a more predominant risk factor in IHD mortality than in CED. We hypothesize that vegetable and artichoke consumption may be less protective for psychosocial stress than for CED mortality, therefore, this could explain the non-significant effect observed on IHD mortality [42].

Regarding the effects of pesticides and fertilizers on CVD mortality [43], the ecological (organic) agriculture was small in the Castellón province during the 1991–2011 period; the median of hectares with ecological agriculture of irrigated cultivation of vegetables was 18.33, and this number represented 1.1% of the median of 1740 hectares of irrigated cultivation of vegetables in the eight municipalities during the same period [44,45].

In our DAG approach, we did not consider factors that were not related to agricultural production. However, in our multilevel models, we studied other factors, considering their relation to CED mortality and vegetable and artichoke production, such as unemployment, particulate matter (PM10) and nitrogen dioxide (NO2) levels, coastal proximity, and drinking water hardness. In all models, direction and significance were maintained for these factors. Water hardness, air pollution indicators (PM10 and NO2), and unemployment produced a decline in the protecting effects of the production of artichokes and vegetables. Water hardness and coastal proximity had fewer effects on the estimates. Overall, these results suggest that vegetable and artichoke production are protective factors with independence of some known factors of CVD mortality.

In ecological studies, CVD mortality (CED, IHD, and hypertensive disease) has been associated with social inequalities and low socio-economic levels [46,47], and low mortality has been observed in rural non-metropolitan counties with agricultural production in the United States [48]. Changes in agricultural production, food processing, and diet in European Mediterranean countries have been related to an increase in CVD incidence [49]. In our study, all-cause, CVD, and CED mortalities were lower with the presence of vegetable and artichoke production, but we found no changes in mortality related to agricultural or citrus production. The largest decline in CVD and CED mortalities associated with artichokes and vegetables production was found in Benicarló. Lower IHD mortality has been associated with fruit and vegetable consumption [50], but not in our study and others [51]. The eight municipalities in our study had differences in agricultural production which may explain CVD mortality variations. In general, a diet with vegetables and fruits is associated with lower all-cause, CVD, and CED mortalities [52,53,54,55]. Apart from artichokes, vegetable cultivation included salads, cauliflowers, cabbages, tomatoes, and onions. Specifically, the consumption of vegetables, including that of cruciferous vegetables (i.e., cauliflowers and cabbages, among others), has been associated with a decline of all-cause, CVD, and CED mortalities. Physiologically, protective effects associated with vegetables include anti-oxidant and anti-inflammatory actions, inhibition of platelet aggregation, lowering of blood pressure, and glucose and lipid metabolism regulation [18,56,57].

Artichokes have several healthy properties including antioxidant activity, lipid and cholesterol reduction, and a decrease in postprandial glycemic and insulin response [58,59,60]. Components of artichokes which have an effect are caffeic acid derivatives, flavonoids, such as luteolin, cynaroside, scolymoside, and cynarotrioside and sesquiterpenic lactones; artichokes decrease production of cholesterol and endogenous triglycerides and increase their excretion by bile production [21,61,62]. In several clinical trials, reductions in cholesterol levels were found but more studies are needed [63,64,65,66]. In addition, improved hypertension, body mass index, cardio-metabolic parameters, and hepato-protective effects have been observed [67,68,69]. Artichoke extract may be relevant as a therapeutic factor for atherosclerosis due to it lowering the rates of low-density-lipoprotein cholesterol and by its antioxidant, anti-inflammatory, and nitric oxide regulation effects [70,71,72,73,74,75].

We hypothesized that the production of vegetables and artichokes may be a proxy of their consumption in our ecological approach. Considering the high production of vegetables and artichokes in Benicarló, it is plausible that vegetable and artichoke consumption may be more elevated than in the other municipalities due to higher accessibility, lower price, different diet habits related with own vegetable production, and the elevated proportion of people with agricultural occupations. More than 200 farmers of Benicarló, Vinaròs, and neighboring towns work in the cultivation of artichokes. In addition, since 1998, the artichoke of Benicarló is an appellation of origin, and since 2003, it holds a Denomination of Protected Origin of the European Commission. A week of artichoke consumption has been celebrated each year since 1993 where artichoke culinary recipes are presented in many restaurants of Benicarló as a part of a multitudinous set of events with artichokes as the protagonist [76,77]. On the other hand, agricultural lands are well distributed with small exploitations, which have a mean of 5.47 hectares per owner in the CV with a Gini index of 0.45 [78]. In the eight municipalities, the median of farm surface per owner was 3.70 hectares (rank 2.06 Castelló de la Plana, 9.97 Benicarló). The health effects of gardening have been indicated and Benicarló has a high agricultural occupation percentage [79].

The study limitations include its ecological design with aggregate data susceptible to ecological fallacy. In addition, other potential confounders related with CVD mortality, such as existing co-morbidities and family history of CVD of this population, were not studied, which we could consider another limitation of our study. However, we obtained similar results with the AMR sensitivity analysis. We assumed that the production of vegetables and artichokes was a proxy of their consumption but studies to verify this point are necessary. Our retrospective approach could increase potential bias; there were few socio-economic data at a municipal level; some residual confounding could affect our estimates, some differences among municipalities may persist, genetic predisposition to CVD of the population [80] was not studied, and finally, we cannot discard the impact of unmeasured or unknown factors.

A strength of the study is that CVD was estimated by SMRs and AMRs in a substantial twenty-year period; the study encompassed relatively small areas with similar climates and small socio-economic differences, and we used multilevel models to adjust for variables considered to be potential confounders and other unknown municipality related effects.

Our new research proposal includes a cross-sectional survey of a representative sample in each municipality to obtain a background of CVD risk factors and co-morbidities, jointly with food consumption, diet, habits, and demographic characteristics as a first step. The survey’s results could support the relationship between the consumption of vegetables and artichokes and CVD and CED mortalities, and it could facilitate the rationale for accurate preventive strategies. As a second step, we recommend a properly powered cohort approach with sufficient follow-up on a population-based study of the first step.

## 6. Conclusions

Our study suggests that the production of vegetables and artichokes may be a protective factor of CED and CVD mortality. Our recommendation is to continue this research using a cohort approach in the eight municipalities.

## Figures and Tables

**Figure 1 ijerph-17-06583-f001:**
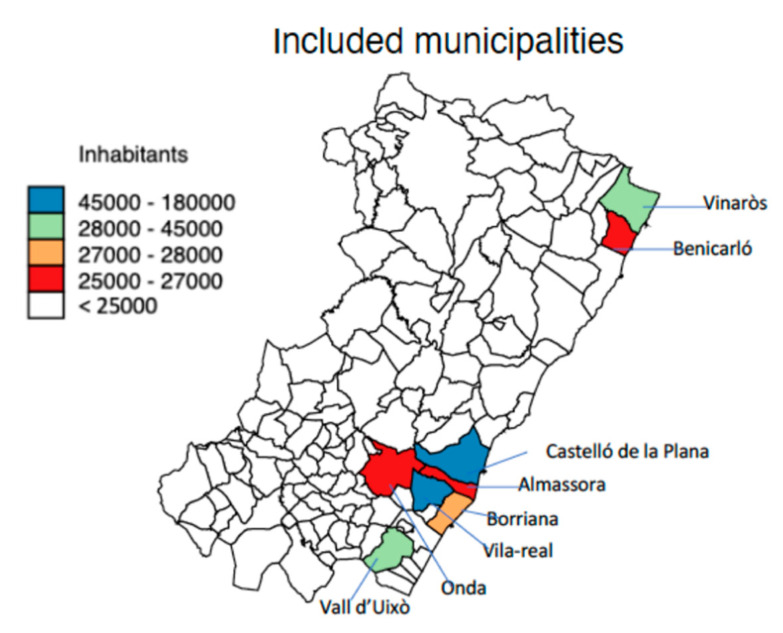
Geographical locations and populations of the included municipalities.

**Figure 2 ijerph-17-06583-f002:**
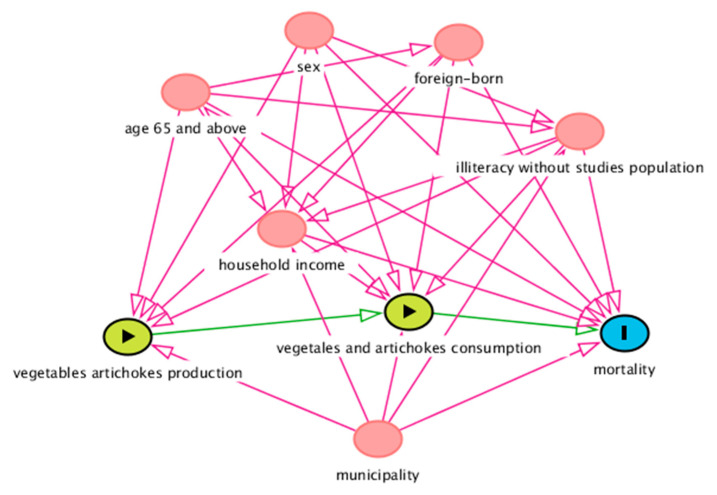
Directed Acyclic Graph (DAG) of the effect of vegetable and artichoke production and consumption (exposure) on mortality (outcome). Ancestors of exposure and outcome (red color). Based on DAGitty version 3.0. ( Johannes Textor, Nijmegen, The Netherlands).

**Table 1 ijerph-17-06583-t001:** Socio-economic characteristics of the eight municipalities, agricultural production in hectares per 1000 inhabitants, and environmental indicators (Castellón, 1991–2011 period).

Variables	Almassora	Benicarló	Borriana	Castelló	Onda	Vall d’Uixó	Vila- Real	Vinaròs
**Socio-economic indicators**								
**Health departments**	2	1	3	2	3	3	3	1
**Population ^1^**	17,384	21,015	27,385	150,444	20,021	29,638	42,961	22,960
**Population aged 65 years and above %**	14.2	17.8	17.1	17.8	15.7	19.1	15.8	18.1
**Foreign-born %**	14.8	21.2	15.9	15.9	12.5	7.1	11.8	17.8
**Household income in euros**	14,064	13,824	14,077	15,519	13,815	13,754	14,436	14,198
**Unemployment %**	18.1	15.3	16.8	17.9	21.2	18.67	19.6	15.3
**Illiteracy %**	32.1	22.6	22.4	20.4	23.3	24.0	24.4	22.9
**Occupation %**								
**Agriculture %**	3.2	8.8	10.0	2.7	5.9	8.6	4.8	8.7
**Industrial %**	22.1	18.7	22.7	17.2	33.1	20.5	26.6	16.7
**Construction %**	11.8	10.1	7.8	6.6	8.7	9.3	8.6	8.2
**Services %**	62.9	62.4	59.5	73.5	52.3	61.7	60.0	66.4
**Agriculture production Ha ^2^**								
**Agriculture ^2^**	104.3	181.0	130.7	46.0	404.0	189.1	88.8	335.9
**Herbaceous ^2^**	8.7	58.9	4.7	7.5	14.6	4.3	4.7	37.0
**Woody plants ^2^**	90.0	105.8	122.9	29.2	242.2	77.9	80.2	281.5
**Pastures ^2^**	5.8	15.8	3.1	8.1	69.9	106.0	3.9	16.3
**Forest species ^2^**	0.15	1.2	0.01	1.21	89.4	0.02	0.0	1.04
**Vegetables ^3^**	1.1	64.0	0.9	0.2	0.2	0.3	0.0	10.3
**Artichoke ^3^**	0.5	27.4	0.2	0.1	0.0	0.03	0.0	2.8
**Citrus ^3^**	94.4	52.3	111.7	21.7	182.7	70.9	76.2	204.6
**Environmental indicators**								
**Water hardness in French degrees ^4^**	35.8°	35.5°	45.8°	49.5°	48.9°	28.9°	45.0°	30.4°
**Coastal distance in km**	5.0	0.10	5.7	5.0	10.6	20.0	8.4	0.10
**PM10 mg/m^3 5^**	36.58	16.03	27.38	29.08	21.93	19.0 ^6^	31.64	16.03
**NO2 mg/m^3 5^**	22.7	6.25	16.22	21.49	9.0	6.5 ^6^	19.20	6.25

^1^ Median population (1991–2011). ^2^ Land in hectares per 1000 inhabitants: Census 2011. ^3^ Median hectare production (2002–2011) per 1000 inhabitants. Portal estadístico Generalitat Valenciana [25]. ^4^ Annual mean values. Sources: Facsa and Sorea water distribution companies www.facsa.com
www.sorea.cat. ^5^ Annual mean concentration (2001–2011). Source: Calidad Ambiental. Generalitat Valenciana [24]. Annual mean concentration (2003–2011). ^6^ Vall d’Uixó (2010–2011).

**Table 2 ijerph-17-06583-t002:** Spearman correlation coefficients among independent variables (Castellón, 1991–2011 period).

Variables	(a)	(b)	(c)	(d)	(e)	(f)	(g)	(h)	(i)	(j)	(k)	(l)	(m)
**Population aged 65 years and above % (a)**	1.0												
**Household income in euros (b)**	−0.06	1.0											
**Foreign-born % (c)**	0.16	0.38	1.0										
**Illiteracy % (d)**	−0.41	−0.33	**−0.71**	1.0									
**Agriculture** **Production ^1^ (e)**	0.22	**−0.69**	−0.17	0.10	1.0								
**Vegetables ^1^ (f)**	0.22	0.33	*0.52*	0.10	0.40	1.0							
**Artichoke ^1^ (g)**	0.25	0.04	**0.75**	−0.29	0.05	**0.91**	1.0						
**Citrus ^1^ (h)**	−0.25	−0.12	−0.22	0.24	*0.62*	0.21	−0.01	1.0					
**Unemployment % (i)**	−0.44	**−0.57**	**−0.83**	*−0.57*	0.05	−0.43	**−0.92**	0.01	1.0				
**Coastal distance in km (j)**	−0.11	−0.42	**−0.88**	0.36	0.14	**−0.66**	**0.87**	0.02	**0.84**	1.0			
**PM10 mg/m^3^/(k)**	**−0.70**	0.40	−0.31	0.37	**−0.72**	*−0.56*	−0.41	−0.14	0.49	0.25	1.0		
**NO2 mg/m^3^ (l)**	**−0.64**	0.42	−0.19	0.23	**−0.74**	*−0.54*	−0.35	−0.22	0.42	0.19	**0.98**	1.0	
**Water hardness ^2^ (m)**	*−0.54*	0.45	0.19	−0.40	0.40	*−0.57*	−0.41	−0.10	0.29	0.11	*0.51*	*0.59*	1.0

^1^ Land in hectares per 1000 inhabitants. ^2^ French degrees. Italic: *p*-value < 0.05; Bold: *p*-value < 0.01.

**Table 3 ijerph-17-06583-t003:** Smoothed standardized mortality ratios (SMRs): males and females (Castellón, 1991–2011 period).

Municipality	All-Cause Mortality		Cardiovascular Disease (CVD)		Cerebrovascular Disease (CED)		Ischaemic Heart Disease (IHD)		CED + IHD	
	Males	Females	Males	Females	Males	Females	Males	Females	Males	Females
**Almassora**	101.2	103.8	99.2	115.5	98.2	98.5	110.2	101.1	103.8	99.1
**Benicarló**	104.7	111.8	94.9	97.7	75.5	71.4	94.1	82.1	83.3	73.9
**Borriana**	95.4	101.0	120.6	116.3	137.0	141.6	113.4	97.6	126.5	129.3
**Castelló**	98.1	107.1	106.4	106.5	100.6	98.9	108.1	117.9	104.1	104.3
**Onda**	115.2	109.1	111.3	120.2	119.3	133.1	107.2	87.0	104.1	120.1
**Vall d’Uixó**	108.2	111.7	105.0	116.1	103.8	106.4	107.2	88.2	114.0	101.0
**Vila-real**	104.1	106.2	113.7	121.0	128.3	138.4	101.3	96.2	115.9	126.4
**Vinaròs**	100.7	103.0	100.6	102.0	85.4	101.7	109.0	90.9	96.3	98.5

**Table 4 ijerph-17-06583-t004:** Age-adjusted mortality rates per 100,000 inhabitants: males and females (Castellón, 1991–2011 period).

Municipality	All-Cause Mortality		Cardiovascular Disease (CVD)		Cerebrovascular Disease (CED)		Ischaemic Heart Disease (IHD)		CED + IHD	
	Males	Females	Males	Females	Males	Females	Males	Females	Males	Females
**Almassora**	952.2	866.2	287.7	382.5	80.8	109.8	80.2	46.4	161.1	156.1
**Benicarló**	891.9	812.0	274.8	335.5	60.7	78.5	68.0	36.2	128.7	114.7
**Borriana**	1082.9	894.8	350.8	399.0	114.7	159.2	82.5	44.8	197.2	203.4
**Castelló**	976.5	834.2	378.3	365.8	83.5	110.9	78.5	53.2	162.0	164.1
**Onda**	977.7	929.0	323.0	414.4	99.2	151.5	77.6	38.5	176.7	190.0
**Vall d’Uixó**	948.0	885.3	303.7	399.0	85.6	118.6	71.2	39.3	159.9	157.7
**Vila-real**	1027.0	925.3	331.2	414.1	106.9	155.7	73.2	43.8	180.1	198.8
**Vinaròs**	996.4	853.6	292.4	350.2	69.5	113.7	79.5	41.0	148.9	154.6

**Table 5 ijerph-17-06583-t005:** Spatial autocorrelation following Moran’s I index of the smoothed standardized mortality ratios (SMRs) and age-adjusted mortality rates (AMRs) of the eight municipalities (Castellón, 1991–2011 period).

SMRs	Moran’s I Index	Z	*p*-Value
**All-cause mortality**	−0.013	0.620	0.268
**Cardiovascular (CVD) mortality**	0.110	2.015	0.022
**Cerebrovascular (CED) mortality**	0.073	1.596	0.055
**Ischaemic heart disease (IHD) mortality**	−0.008	0.741	0.229
**CED + IHD mortality**	0.110	2.052	0.020
**AMRs**			
**All-cause mortality**	0.006	0.699	0.242
**Cardiovascular (CVD) mortality**	0.046	1.292	0.098
**Cerebrovascular (CED) mortality**	0.073	1.199	0.115
**Ischaemic heart disease (IHD) mortality**	0.006	0.817	0.207
**CED + IHD mortality**	0.113	2.083	0.019

**Table 6 ijerph-17-06583-t006:** Multilevel linear regression: smoothed standardized mortality ratio (SMR) and age-adjusted mortality rate (AMR) of mortality among eight municipalities of the Castellón province. Adjusted models for sex, population aged 65 years and above, foreign-born, household income, illiteracy level, and municipality (Castellón, 1991–2011 period).

Factors	SMRs			AMR		
	RC ^1^(95% CI ^2^)	*p*-Value	ICC ^3^ (95% CI)	RC ^1^ (95% CI ^2^)	*p*-Value	ICC ^3^ (95% IC)
**All-cause mortality**						
**Agriculture ^4^**	−0.01 (−0.04 to 0.43)	0.726	0.24(0.01 to 0.92) ^5^	−0.04 (−0.27 to 0.20)	0.761	0.25 (0.00–0.91)
**Vegetables ^4^**	−0.19 (−0.31 to −0.07)	0.003	NC ^6^	−1.76 (−2.88 to −0.63)	0.002	NC ^6^
**Artichokes ^4^**	−0.42 (−0.62 to −0.11)	0.003	NC ^6^	−3.92 (−6.41 to −1.42)	0.002	NC ^6^
**Citrus ^4^**	0.03 (−0.01 to 0.07)	0.196	0.16 (0.00 to 0.97) ^5^	0.25 (−0.11 to 0.61)	0.178	0.15 (0.00 to 0.97)
**Cardiovascular disease (CVD) mortality**						
**Agriculture ^4^**	−0.02 (−0.05 to 0.01)	0.169	0.23 (0.01 to 0.92)	−0.05 (−0.17 to 0.06)	0.376	NC ^6^
**Vegetables ^4^**	−0.20 (−0.36 to −0.03)	0.017	0.07(0.00 to 0.99)	−0.58 (−1.33 to 0.16)	0.126	NC ^6^
**Artichokes ^4^**	−0.42 (−0.79 to −0.04)	0.028	0.10 (0.00 to 0.99)	−1.20 (−2.88 to 0.48)	0.161	NC ^6^
**Citrus ^4^**	0.02 (−0.01 to 0.06)	0.137	0.88 (0.50 to 0.98)	−0.02 (−0.22 to 0.18)	0.862	NC ^6^
**Cerebrovascular disease (CED) mortality**						
**Agriculture ^4^**	−0.05 (−0.14 to 0.04)	0.309	0.82 (0.54 to 0.96)	−0.04 (−0.14 to 0.05)	0.372	0.62 (0.21 to 0.90)
**Vegetables ^4^**	−0.68 (−1.16 to −0.19)	0.006	0.72 (0.33 to 0.93)	−0.71 (−1.18 to −0.24)	0.003	0.39 (0.05 to 0.88)
**Artichokes ^4^**	−1.47 (−2.57 to −0.36)	0.000	0.73 (0.35 to 0.93)	−1.54 (−2.16 to −0.47)	0.005	0.41 (0.06 to 0.88)
**Citrus ^4^**	0.04 (−0.13 to 0.21)	0.623	0.84 (0.54 to 0.96) ^5^	0.05 (−0.12 to 0.22)	0.544	0.64 (0.23 to 0.99)
**Ischaemic heart disease (IHD) mortality**						
**Agriculture ^4^**	0.01 (−0.05 to 0.07)	0.771	NC ^6^	0.01 (−0.02 to 0.03)	0.651	0.19 (0.0 to 0.95)
**Vegetables ^4^**	−0.36 (−0.76 to 0.03)	0.067	NC ^6^	−0.21 (−0.31 to −0.12)	0.000	NC ^6^
**Artichokes ^4^**	−0.78 (−1.65 to 0.10)	0.083	NC ^6^	−0.46 (−0.68 to −0.25)	0.000	NC ^6^
**Citrus ^4^**	0.30 (−0.07 to 0.14)	0.548	NC ^6^	0.03 (−0.01 to 0.06)	0.104	0.06 (0.00 to 0.99)
**Cerebrovascular disease (CED) and ischaemic heart disease (IHD) mortality**						
**Agriculture ^4^**	−0.02 (−0.10 to 0.05)	0.509	0.80 (0.47 to 0.95) ^5^	−0.04 (0.15 to 0.08)	0.528	0.80 (0.45 to 0.95) ^5^
**Vegetables ^4^**	−0.57 (−0.89 to −0.26)	0.000	0.59 (0.18 to 0.90)	−0.92 (−1.41 to −0.42)	0.000	0.56(0.16 to 0.90)
**Artichokes ^4^**	−1.24 (−1.97 to −0.52)	0.001	0.61 (0.20 to 0.91)	−2.00 (−3.14 to −0.86)	0.001	0.58 (0.17 to 0.90)
**Citrus ^4^**	0.05 (−0.07 to 0.17)	0.434	0.80 (0.47 to 0.95) ^5^	0.08 (−0.12 to 0.28)	0.421	0.79 (0.44 to 0.95) ^5^

^1^ RC = Regression coefficient ^2^ CI = confidence interval. ^3^ ICC= interclass correlation ^4^ Hectare production per 1000 inhabitants. ^5^ Likelihood ratio test (LRT) value > 0.05 ^6^ NC = Not computable.

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
