# Peer review of "Production of Vegetables and Artichokes Is Associated with Lower Cardiovascular Mortality: An Ecological Study"

_ijerph, 2020, doi:10.3390/ijerph17186583_

Round 1

Reviewer 1 Report

In this manuscript from Arnedo-Pena and colleagues, the authors tried to establish a connection between production of vegetables and artichokes and cardiovascular mortality in several municipalities of Spain. The idea at the basis of this study is that the production of these foods may be a proxy for health behaviours (specifically, consumptions of vegetables and artichokes), this in turn influencing the mortality of the inhabitants of these regions.

Although the authors deserve credits for the statistical methodology through which they tried to analyze these data, I am concerned that the logical assumption at the basis of this study may not be reliable. In fact, even if the production of vegetables and other foods may be an indirect indicator for the local consumption, this is not a given and should be demonstrated on a case-by-case basis.

Also, there are several other potential confounding factors that were not considered in this study, some of which may be somewhat linked with the production of vegetables and artichokes, including existing comorbidities, familiar history of CVD etc. The adjustement for the so-called "gender-related variables" (household income, educational level etc.) is right and the authors should be commended for this, but the lack of adjustement for other crucial variables, even if unavaidoable according to the design of the study, severely limit the significance of this manuscript.

In conclusion, even if the idea at the basis of this paper may be interesting and sounding, I do not think that reliable connection between production of vegetables and artichokes and CVD are demonstrated by this study, given the issues mentioned above.

Other issues:
- Please refer correctly to term "gender" and "sex". When the authors refers to the biological sex (i.e. male vs. female), they should use the term Sex instead of Gender (please refer to 10.1001/jama.2016.16405)
- There are few typos across the manuscript (e.g. row 98: "lineal" instead of "linear").

Reviewer 2 Report

The authors investigated the relationship between production of vegetables and artichokes and cardiovascular mortality in some areas of Spain. They showed that the production of vegetables and artichokes were associated with lower rates of cerebrovascular disease (CED) and cardiovascular disease (CVD). This study seems to be unique and I had to pay respect to much effort of collecting data, however, there have been several problems to be solved.

Major comments.                                                                                      #1 The authors collected and adopted socio-economic and demographic data in the present study. However, there have been a lot of factors affecting the occurrence of ischaemic heart disease (IHD) or CED, such as the frequencies of conventional risk factors of smoking, hypertension, dyslipidemia and diabetes mellitus. Furthermore, the previous medical history, such as IHD or taking statins, may affect the occurrence of IHD, CED, or CVD. The authors had better show and adopt such data about risk factors and/or medical information in the present study.                                                                                    

#2 According to Figure 1, Benicarlo and Vinaros were apart from other areas. In the present study, as for me, data of Benicarlo seemed to affect the all results of present study. The authors should comment on this and why the authors adopted these two apart areas.                                                                                                                                                                            #3 In the "Discussion" section, the authors speculated the elevated consumption of vegetables and artichokes as the result of close relationship between production of vegetables and artichokes and low frequencies of CED and CVD. The authors had better speculate the responsible mechanisms for this main result. In addition, the authors had better show any data of elevated consumption of vegetables and antichokes in two areas.                                                                                                                                                    #4 The authors had better comment on the significant relationship between production of vegetables and artichokes and reduced frequency of CVD, not but of IHD.                                                                                                                                                                                                                Minor comments                                                                                          #1 In the "Abstract" section, IHD was an abbreviation of ischaemic heart disease. In addition, IHD did not seem to be shown as a full-spell notation in the main text. 

Reviewer 3 Report

In the present work (Production of vegetables and artichokes and 2 cardiovascular mortality: an ecological study), authors expose the protective influence of vegetable production (specially artichokes) in cardiovascular diseases (CVD). Authors analyzed 8 municipalities with more than 25000 inhabitants in Castellon´s province. This have different incidence in CVD and different vegetable production, which lead to the hypothesis, the influence of vegetable production as a protective factor for CVD. Authors carried out a multilevel lineal regression to estimate associations between mortality outcome and exposure variables in models adjusted for potential confounders.

Present work exposes an interesting idea. Although working hypothesis is well planned and results seem interesting, there are some concerns about how it is carried out and more data may be included to improve the work.

Concerns:

1) Authors carried out a multilevel lineal regression adjusted for potential confounders. Since there are few differences among municipalities, these potential confounders included in analysis should be at least mentioned, to confirm that this differences among municipalities are controlled in some way.

2) A new table, with Multilevel linear regression for Standardized Mortality Ratio smoothed (SMRs) and Age-Adjusted Mortality Rate (AMR) (similar to table 6), but including all confounders included in analysis, should be included (at least as supplemental material, if no influence in SMRs and AMR is found).

3) In discussion authors mention the potential influence of different external factors in CVD incidence. They also mention as limitation the ecologic design with aggregate data. In this sense, although multilevel regression is adjusted for potential confounders, they should also mention the significant differences among municipalities as other potential limitation.

4) In line 104 it seems that there is a spelling mistake: “Spearman no (non) parametric correlations to detect relationships among exposure variables”.

5) Authors include for analysis 8 municipalities with more than 25000 inhabitants in Castellon´s province. I suppose this are the only 8 municipalities with more than this number of inhabitants, but this is not mentioned in the text. If I am correct, authors should address that this are the only municipalities with more than 25000 inhabitants.

Author Response

In the present work (Production of vegetables and artichokes and 2 cardiovascular mortality: an ecological study), authors expose the protective influence of vegetable production (specially artichokes) in cardiovascular diseases (CVD). Authors analyzed 8 municipalities with more than 25000 inhabitants in Castellon´s province. This have different incidence in CVD and different vegetablye production, which lead to the hypothesis, the influence of vegetable production as a protective factor for CVD. Authors carried out a multilevel lineal regression to estimate associations between mortality outcome and exposure variables in models adjusted for potential confounders.

Present work exposes an interesting idea. Although working hypothesis is well planned and results seem interesting, there are some concerns about how it is carried out and more data may be included to improve the work.

Thank you very much for reviewing our manuscript. We value your concerns and we have tried to follow your indications.

Concerns:

1) Authors carried out a multilevel lineal regression adjusted for potential confounders. Since there are few differences among municipalities, these potential confounders included in analysis should be at least mentioned, to confirm that this differences among municipalities are controlled in some way.

The potential confounders were studied following the Direct Acyclic Graphs (DAGs) approach and were mentioned in the manuscript (Table 6  ). Adjusted models for gender, population age 65 and above, foreign-born, household income, and illiteracy-without studies, and we now mention the potential confounders in Material and Methods, additionally in the discussion, we mention other potential confounders.

2) A new table, with Multilevel linear regression for Standardized Mortality Ratio smoothed (SMRs) and Age-Adjusted Mortality Rate (AMR) (similar to table 6), but including all confounders included in analysis, should be included (at least as supplemental material, if no influence in SMRs and AMR is found).

The exposure variables include agriculture, vegetables, artichokes and citrus productions and the potential confounders were studied by DAG and we added other potential confounders such as unemployment, water hardness, coastal distance, and air pollution (PM10 NO2) with relationship to cardiovascular diseases (CVD), and we mentioned that the protection of vegetables and artichokes production was maintained. The DAG approach indicates that each exposure variable has a conjunct of potential confounders and it needs be analyzed one by one (1). For example, air pollution may be associated with CVD mortality in the 8 municipalities but a DAG exclusive for this exposure is necessary. A table with the results of all confounders studied jointly may have a problem of Table 2 fallacy (2). In addition, our hypothesis is only on vegetables and artichokes production and CVD mortality.              

3) In discussion authors mention the potential influence of different external factors in CVD incidence. They also mention as limitation the ecologic design with aggregate data. In this sense, although multilevel regression is adjusted for potential confounders, they should also mention the significant differences among municipalities as other potential limitation.

We agree with the review that other differences among municipalities may co-existent and it is a limitation of the study. It is included in the manuscript.

4) In line 104 it seems that there is a spelling mistake: “Spearman no (non) parametric correlations to detect relationships among exposure variables”.

Thanks for your indication; we have amended the mistyping in the manuscript.

5) Authors include for analysis 8 municipalities with more than 25000 inhabitants in Castellon´s province. I suppose this are the only 8 municipalities with more than this number of inhabitants, but this is not mentioned in the text. If I am correct, authors should address that this are the only municipalities with more than 25000 inhabitants.

Certainly, we studied all municipalities with 25,000 or more inhabitants of Castellón´s province. We now make this clear in the manuscript.

References

  1. Lederer DJ, Bell SC, Branson RD, et al. Control of confounding and reporting of results in causal inference studies. Guidance for Authors from Editors of Respiratory, Sleep, and Critical Care Journals. Ann Am Thorac Soc. 2019;16:283. Ann Am Thorac Soc. 2019;16(1):22-28.
  2. Westreich D, Greenland S. The table 2 fallacy: presenting and interpreting confounder and modifier coefficients. Am J Epidemiol. 2013;177: 292-8.

Reviewer 4 Report

The study entitled” Production of vegetables and artichokes and cardiovascular mortality: an ecological study” by Arnedo-Pena et al described the relationship between vegetable and artichoke production with mortality rate of CVD. This study is well established and designed and I have some comments to improve the quality of MS and to be clear for readers.

  1. The authors mentioned the effects of vegetables productions on CVDs however, I suggest to add a table containing the names of vegetables produced in this regions, their annual production, the main phytochemical constituents and their pharmacodynamic effect (ass this table in the discussion part.
  2. The title has to be more precise.
  3. How the authors know that the decrease of MR related to the consumption of these vegetables as well as many regions of Spain consume high amounts of Olive oil or they have an healthy life style.
  4. genes and specific DNA sequence variants responsible for CVD, Forthat the people in this region of Spain have the genes and DNA sequences that protect them from CVD. Kindly read this study (Kathiresan, Sekar, and Deepak Srivastava. "Genetics of human cardiovascular disease." Cell6 (2012): 1242-1257.)
  5. Kindly, is the production of different vegetables in the study area organic or not? The authors have to mention the type of agriculture in this area because some pesticides and fertilizers in agriculture may play negative effect on human health including CVD.
  6. The Conclusion section must contain the outcomes of your study briefly also your recommendations. Therefore, kindly rewrite the conclusion section to contain minimum two paragraphs.
  7. Correct the reference number 2.
  8. Many sentences have to be rephrased and the MS needs grammatical, typos and editing corrections.

Round 2

Reviewer 1 Report

In this revised version of the manuscript from Arnedo-Pena and colleagues, the authors answered part of my previous comments, raised in the first round of review. 
I think that previous issues regarding the reliability of the relationship investigated in this study still remain, and represent the major limitation of this article. The authors may, at least, consider to strongly and extensively address this as a major limitation of their study (currently, they only state that "The study limitations include its ecologic design with aggregate data susceptible to ecologic fallacy [...]", along with providing some insights of how further studies should be designed to overcome this issues. 
A clear definition of the issues reported in my previous round of review may be useful to - at least - counterbalance the discussion and the interpretation of this manuscript.

Author Response

Response to the first reviewer

In this revised version of the manuscript from Arnedo-Pena and colleagues, the authors answered part of my previous comments, raised in the first round of review. 
I think that previous issues regarding the reliability of the relationship investigated in this study still remain, and represent the major limitation of this article. The authors may, at least, consider to strongly and extensively address this as a major limitation of their study (currently, they only state that "The study limitations include its ecologic design with aggregate data susceptible to ecologic fallacy [...]", along with providing some insights of how further studies should be designed to overcome this issues.  
A clear definition of the issues reported in my previous round of review may be useful to - at least - counterbalance the discussion and the interpretation of this manuscript.

Thank you very much for your second review of our manuscript.

We agree with your comments and include the responses, the first on study’s limitations and the second on further research, in the manuscript.

Reviewer 2 Report

The authors revised their manuscript appropriately according to the reviewers' comments. Now I have no request ant question regarding the revised manuscript.

Author Response

Thank you very much for your second review of our manuscript. 

Reviewer 4 Report

The authors carried out all the required corrections 

Author Response

Thank you very much for your second review of our manuscript

Round 3

Reviewer 1 Report

Thank you for the opportunity to review another round of review of this manuscript. 
In this version the authors expressed more clearly the limitations of their study and I think the discussion is more balanced than in previous rounds.